

# A PCA-based bio-motion generator to synthesize new patterns of human running

José María Baydal-Bertomeu[*], Juan Vicente Durá-Gil[*], Ana Piérola-Orcero[*], Eduardo Parrilla Bernabé[*], Alfredo Ballester[*] and Sandra Alemany-Munt[*]

Instituto de Biomecánica de Valencia, Universitat Politècnica de València, Valencia, Spain
[*] These authors contributed equally to this work.

## ABSTRACT

Synthesizing human movement is useful for most applications where the use of avatars is required. These movements should be as realistic as possible and thus must take into account anthropometric characteristics (weight, height, etc.), gender, and the performance of the activity being developed. The aim of this study is to develop a new methodology based on the combination of principal component analysis and partial least squares regression model that can generate realistic motion from a set of data (gender, anthropometry and performance). A total of 18 volunteer runners have participated in the study. The joint angles of the main body joints were recorded in an experimental study using 3D motion tracking technology. A five-step methodology has been employed to develop a model capable of generating a realistic running motion. The described model has been validated for running motion, showing a highly realistic motion which fits properly with the real movements measured. The described methodology could be applied to synthesize any type of motion: walking, going up and down stairs, etc. In future work, we want to integrate the motion in realistic body shapes, generated with a similar methodology and from the same simple original data.

Corresponding author
Juan Vicente Durá-Gil,
juan.dura@ibv.upv.es

## INTRODUCTION

It is well known that there is a large degree of information contained in the kinematics of a moving body which is influenced by parameters such as: gender, age, anthropometrical features, emotional state, personality traits, etc. (*Troje, 2008*). A number of studies demonstrate the capability of the human visual system to detect, recognize and interpret the information encoded in the biological motion (*Johansson, 1973*). There are also many attempts to analyse this information encrypted in human motion. Some researchers use discrete kinematics parameters such as ranges, speed, etc. (*Dvorak et al., 1992*). Others focus their studies on the sequence of movement along time instead of recording simple parameters. In these cases, they analyse the complete function of time $f(t)$ (*Feipel et al., 1999*). A number of kinematical models are based on frequency domain manipulations (*Davis, Bobick & Richards, 2000*) and multiresolution filtering (*Bruderlin & Williams, 1995*).

Nevertheless, the most common objective of these studies is to model and to classify the movement pattern of the person being measured, rather than creating new motions from the extracted information.

In this regard, motion synthesis is currently attracting a great deal of attention within the computer graphics community as a means of animating three dimensional realistic characters and avatars; and in the robotic field to provide controlled real-time dynamic motion for the locomotion and other activities (*Kajita et al., 2002*). With the computational resources available today, large-scale models of the body (i.e., models that have many degrees of freedom and are actuated by many muscles) may be used to perform realistic simulations (*Pandy, 2001*). Nevertheless, it is necessary to perform lab experiments to track the positions and orientations of body segments executing the task aimed to be synthesized. Recording motion data directly from real actors and mapping them to computer characters is a common technique used to generate high quality motion (*Li, Wang & Shum, 2002*). However, this technique requires a high effort in experimental work. Besides, new measures are needed to include changes in the pattern of movement, such as age, weight, gender or speed. In this sense, it would be useful to create a methodology based on biomechanical models constructed from a database of motions, instead of a single actor, able to generate realistic motions of individuals with different anthropometrical characteristics, with sufficient accuracy and without the need to perform laboratory measurements.

Several authors have addressed the motion modelling and synthesis for biped walking, jumping, pedalling (*Troje, 2002*), or even stair-ascending (*Vasilescu, 2002*). Classically the mathematical approach of the synthesis of movement has been the dynamic optimization of biomechanical body structures (*Pandy, 2001*). These models provide detailed information of the functioning of some structures, such as the description of muscle function during normal gait. However, this approach becomes an unworkable problem when a greater number of body structures are included in the model. A new approach based on Principal Components Analysis (PCA) can facilitate the understanding of the information contained in the kinematics of a moving human body and avoids the inclusion of the dynamics in the model. PCA can extract depth information contained in the mathematical function and its derivatives not normally available through traditional statistical methods (*Ullah & Finch, 2013*). In this way PCA can be used on different levels. For instance, *Troje (2002)* used PCA in two steps for the purpose of analysing and synthesizing human gait patterns. In the first one, they extracted the main components from the entire database, in order to eliminate redundancy and to reduce the dimensionality. In the second step, PCA was applied particularly for each walker in order to retain the encoded information of each walker-specific variation. In our research, we will also use a model (based on PCA) to extract the most relevant information from the pattern of running. This information will be used to develop a bio-motion generator which will solve the opposite problem of synthesizing new realistic movements.

In addition, existing literature focused on synthesizing motion does not correlate the generated movement to age, gender, performance parameters such as velocity or anthropometrical features. In this sense our research has three goals. The first one is to generate a database of running movements of a full human model. The second is to

extract the signature of each motion, by means of PCA technique and to correlate the distinctive styles of each runner with their anthropometrical characteristics, age, gender, and performance parameters such as the velocity of the action. The third is to develop a bio-motion generator based on a statistical model capable of synthesizing new realistic running motion from a set of desired data: age, gender, height, body mass index (BMI) and velocity.

Nowadays there exists a line of research developed in the field of anthropometry for the purpose of obtaining a model of human body shape from a database of processed raw scans (*Vinué et al., 2014*). The methodology followed in that line of research provides sufficient resolution to synthesize accurate realistic representations of body shapes from a set of simple anthropometrical parameters. *Ballester et al. (2014)* describe a method based on the harmonization of body scan data followed by a Shape Analysis procedure using Principal Component Analysis. The combination of these techniques allows the generation of human 3D shape models from anthropometric measurement data (age, height, weight, BMI, waist girth, hip girth, bust/chest girth, etc.). Our hypothesis is that the use of a similarly based methodology to generate human motion instead of human body shapes is possible, valid and reliable. The novelty in our approach is the generation of running data from a set of easily measurable anthropometric parameters and a desired value of running speed.

## MATERIALS AND METHODS

### Data gathering

An experimental phase was carried out with the aim of gathering a database of the running movements that we needed to include in the biomechanical model. The data consisted of the 3D joint angles of the main body joints.

The articulated human body model used in our study comprised 21 segments and 20 joints distributed throughout the body. Lower limb: hip, knee, ankle and metatarsophalangeal joint; upper limb: shoulder, elbow and wrist; trunk: pelvis, L5, L3, T12, T8 and Neck.

The positions of the joints of the human body model were defined in a recursive mode with respect to the origin joint (father) of the related segment. This methodology is based on the BioVision Hierarchical data (BVH) format (*Meredith & Maddock, 2001*).

Each subject was kinematically characterised by means of 64 variables, defined as follows:
1. Vertical (Z) position of the root segment, in our case the hips.
2. Tri-dimensional orientation (X, Y, Z) of the total amount of segments with respect the root. $21 \times 3 = 63$ variables.

### Study sample

Eighteen people composed the study sample, with the same number of male and female. Their ages ranged from 21 to 44 years (average age: 31 years). They were selected according to some specific parameters, trying to cover a wide range in the anthropometric characteristics of height and body mass index (Table 1). Ethical approval was obtained from the ethics committee of the Universitat Politècnica València. All participants gave written informed consent.

**Table 1  Description of the anthropometrical parameters.**

| Gender | Parameter | Mean | Std. | Max. | Min. |
|---|---|---|---|---|---|
| Male | Age | 32.0 | 5.1 | 40 | 26 |
| | Weight | 86.712 | 16.813 | 118.000 | 59.000 |
| | Height | 1.779 | .086 | 1.910 | 1.630 |
| | BMI | 27.263 | 4.118 | 33.412 | 21.914 |
| Female | Age | 30.5 | 8.472 | 44 | 21 |
| | Weight | 74.596 | 17.535 | 108.000 | 50.000 |
| | Height | 1.675 | .087 | 1.780 | 1.550 |
| | BMI | 26.362 | 4.747 | 34.473 | 19.257 |

## Measurement and protocol

The measurements were performed using commercial equipment based on 17 inertial sensors: MOVEN studio. The commercial system has been validated by previous studies (*Zhang et al., 2013*; *Thies et al., 2007*). A sampling frequency of 120 Hz was used. This system showed a very high sensitivity to electromagnetic fields. For this reason, the measurements of running trials were done outdoors in a location free of electromagnetic pollution.

## Experimental procedure

For the purpose of controlling the pace of running, a 20-metre-long corridor, delimited with cones every 5 m was set up. Thus, we obtained four areas, one area of acceleration, two of constant speed and a final deceleration area. Running at constant speed presents a periodic timing in which the period depends on the velocity (*Novacheck, 1998*). Nevertheless, acceleration and deceleration periods are out of phase and the duration of cycles is variable. Therefore, the running cycles used to create our model were selected within the area of constant speed.

In the case of running, the pattern of the movement changes with velocity (e.g., stride length, maximum joint angles, etc.). For this reason, each runner completed six running trials at different speeds. Initially, subjects started running at normal speed. In the second measurement subjects ran at their maximum speed. The third and fourth trial were performed at a pace between normal and maximum speed. The fifth trial was performed at the minimum speed at which each runner was able to run, on the edge between walking and running. Running defined in this case as when there is no phase of bipodal support (*Biewener et al., 2004*). The last trial was performed at an average speed between the lower and the normal speed.

This procedure allowed us to obtain six observations representing the whole range of speeds that each subject could execute.

## Mathematical procedure

The methodology used in our study comprised five steps:

1-**Reduction of intra-personal variability**: joint angles are periodic by nature. We took the most representative single stride for the purpose of reducing variability and dismiss the phases with no consistent speed, such as acceleration and deceleration steps. The selected stride was picked in the middle of the running sequence, guaranteeing constant speed.

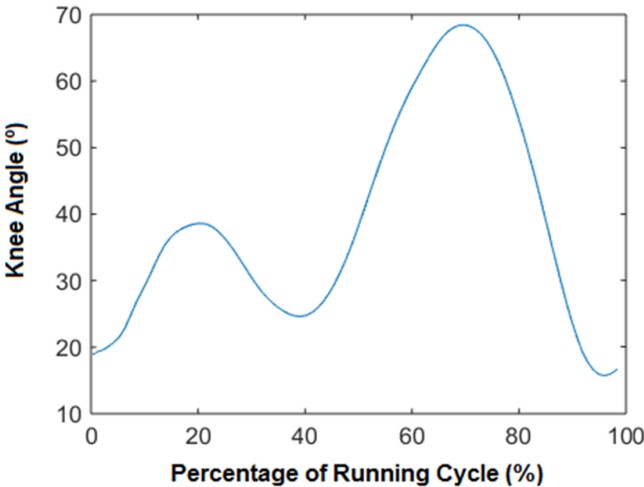

**Figure 1** Knee angle vs % of the cycle.

2-**Time normalization**: time normalization is usually employed for the temporal alignment of cyclic data obtained from different trials with different duration. In our approach, the number of samples for each stride depended on the velocity of the running. At this point we normalised the variable "time" applying an interpolation technique based on cubic splines through the measured values of the whole sequence of samples. This technique enables the normalization of all the measurements to the percentage of the running cycle. The cubic spline was applied to normalize at 50 equispaced time intervals per each variable (see the example in Fig. 1). The application of the cubic spline to the 64 kinematical variables makes a total amount of $50 \times 64$ (3,200) data per each subject.

3-**Data cleaning**: at this point a detailed checking and cleaning of inaccuracies of the kinematical data was conducted. These type of inaccuracies were caused mostly by the measurement system. The prevention of errors at this point is preferable to their later correction once the model has been created. All the measurements have been manually analysed thoroughly by an experienced examiner. The identified inaccuracies were treated as follows:

(a) **Angular offsets:** this common error usually appears during the standing posture and can affect the later joint angles registers during the trial (running) (*Mills et al., 2007*). Offsets have been corrected manually eliminating (adding or subtracting) the difference between the initial angle observed and the expected angle of the body segment at this position.

(b) **Positioning error:** due to the fact that our measurement system, based on inertial sensors, uses the earth's magnetic field to determine the reference position of each subject, it is quite common to find subjects with slight differences in their initial reference positions. In this case, we have proceeded by correcting the reference system aligning it with the direction of running forward. Thus, we can guarantee that all measurements are equally oriented.

(c) **Non-physiological angles:** some errors in the registration of joint angles were detected in the database. These inaccuracies came from errors of the inertial sensors. In this case, it was not possible to correct the error effectively, thus we proceed by eliminating these observations from the database and repeating the measurement.

4-**Dimensionality reduction**: the database of all measurements was combined in a single matrix $W$. The initial number of observations is 108 (18 subjects $\times$ 6 velocities = 108). But three measurements fail, therefore $W$ has 105 rows (observations) and 3,200 columns (50 equispaced time intervals $\times$ 64 kinematical variables). Motion data of each observation is enclosed in the rows of the matrix $W = (w_i), i = 1, \ldots, 105$.

Before the creation of the bio-motion generator, by means of a regression model, it was needed to reduce the dimensionality of the motion data. Computing a PCA on the running data (contained in matrix $W$), resulted in a decomposition of the data matrix $W$ into an average running vector $w_0$ and 3,200 weighed components, arranged in a 3,200 $\times$ 3,200 matrix $V$:

$$W = W_0 + \boldsymbol{\alpha} \cdot V \tag{1}$$

where $W_0$ is a 105 $\times$ 3,200 matrix with all rows equal to $w_0$ and $\boldsymbol{\alpha} = (\alpha_i)$ with $i = 1, \ldots, 105$ is a 105 $\times$ 3,200 matrix of PCA scores. Each observation $w_i$ was thus expressed by a linear combination of scores $\alpha_i$ and PCA components (columns of matrix $V$). Components represented factors related to gender, anthropometrical traits and running speed. And scores represented individual characteristics of each runner and performance of the running trial related to the previous factors. PCA components are arranged in descendent order of explained variance of the original matrix data. Thus the first columns in matrix $V$ retained most of the information in the data sample and it was possible to select a reduced number of components $c$.

$$w_{ic} = w_0 + \alpha_{ic} \cdot V_c. \tag{2}$$

Above $w_0$ denoted the average of all the running samples. The matrix $V_c$ contained the first components. $\alpha_{ic}$ represents the $c$ scores of each observation of the database in the reduced dimension space formed by the selected components. As score values change from negative to positive values, the movement of the runner changes from men to women; high BMI to low BMI; high speed to low speed, etc.

The decision of how many components to retain was a critical issue in the exploratory factor analysis. To perform this decision we used the methodology of Parallel Analysis (PA) (*Hayton, Allen & Scarpello, 2004*). PA is a Monte-Carlo based simulation method that compares the observed eigenvalues (components) with those obtained from randomized normal variables. A component is retained if its explained variance or information is higher than the information provided by the eigenvectors derived from the random data.

5-**Regression model**: one of the objectives of our work was to generate a statistical model capable of synthetizing new realistic running motion from a set of desired data: age, gender, height, weight and velocity, also called 1D data. Accordingly, to devise the

bio-motion generator, we needed to establish the correlation between the 1D data and PCA scores, which provide the signature of each motion.

The correlation was obtained as a regression model, combining a partial least squares (PLS) regression model as a first step and a linear regression model (LRM) as a second step.

PLS methodology is explained in *Wold (2006)* and *Geladi & Kowalski (1986)*. This type of regression model is suitable for the kind of data involved in the bio-motion generator since the input data of the model is strongly correlated (anthropometrical information).

The PLS regression model takes the 1D data—age, height, weight and velocity—as input information and produces a set of PCA scores as output. The LRM model was applied to these output PCA scores to reflect the influence of gender in the PCA scores.

In the first step, we estimated a PLS model considering anthropometrical data and velocity of the movement as independent variables and the PCA scores as dependent variables. The general formula of a PLS model is:

$$Y - Y_0 = B \cdot (X - X_0) + E \tag{3}$$

where $Y$ is the matrix of dependent variables, $X$ is the matrix of independent variables, $X_0$ and $Y_0$ are the matrices of mean data, $B$ is the coefficient matrix of the PLS model and $E$ is the prediction error matrix.

Having four different 1D measurements for each subject $x_i = [age, height, weight, velocity]$, we built a $105 \times 4$ matrix $X$ formed by the concatenation of the vectors $x_i$. Notice that since matrix $= \boldsymbol{\alpha}$ , it is constituted by PCA scores and the mean matrix $Y_0$ is the zero matrix. Finally, we estimated the coefficient matrix $B$ from sample data $X$ and $\boldsymbol{\alpha}$ with the PLS algorithm and substituted in the equation above, obtaining the following model:

$$\boldsymbol{\alpha} = B \cdot (X - X_0) + E. \tag{4}$$

Where $X_0$ is the matrix of mean anthropometrical data and velocity, and $E$ is the prediction error matrix. PLS decomposes the independent and dependent variables in component spaces in order to obtain their correlation. The number of significant PLS components in the model was selected in a leave-one-out procedure and according to the explained variance ($R^2$) criteria.

Secondly, the influence of gender was modelled with a LRM of the prediction error matrix $\boldsymbol{E}$ with coefficients $\boldsymbol{a} = (a_1, \ldots, a_c)$ and $\boldsymbol{b} = (b_1, \ldots, b_c)$, where $c$ is the number of retained PCA components:

$$\hat{E} = \boldsymbol{a} + \boldsymbol{b} \cdot gender \tag{5}$$

where $gender = 0$ for men and $gender = 1$ for women.

This way, the motion information related to gender which is part of the PLS error matrix $E$, and uncorrelated with the prediction derived from the PLS regression, was modelled. Notice that $a_j$ and $b_j$ were considered zero whenever their $F$-value was below a desired level of statistical significance of 95%.

Once we have obtained $B$, $a$ and $b$, whenever we want to synthesize a running motion from new anthropometrical data and velocity, we obtain the corresponding scores $\hat{\alpha}$ of the

new realistic running motion by the following formula:

$$\hat{\alpha} = [a - B \cdot X_0] + \begin{bmatrix} B \\ b \end{bmatrix} \cdot \begin{bmatrix} X & gender \end{bmatrix} \qquad (6)$$

where $X_0$ is the matrix of mean anthropometrical data and velocity.

## Validation methodology of the bio-motion generator

To validate the five-step methodology described to develop the bio-motion generator we propose a comparison between each recorded observation and the prediction of running motion generated by the model by means of the 'leave-one-out' procedure. The recorded observation is considered the true angle curve of the running motion. The predicted motion is estimated using the 'leave-one-out' validation technique; that is, not including that observation in the bio-motion generator. We wish to determine if both curves are reproducible and sufficiently similar to consider that they represent the same motion. For this purpose, we use the Intraclass Correlation Coefficient (ICC) as a measurement of the reliability, and the Standard Error of Measurement (SEM) as a direct measurement of the global error between true and predicted angles. Theoretically, the ICC is defined as the ratio between the true variance and the predicted variance. The ICC varies between 0 and 1 and can be interpreted as the proportion of variance due to the methodology (true versus predicted data) in the total variance. An ICC greater than 0.8 is generally considered to be good (*Fleiss, 1999*). The ICC is determined between the measured or true curve ($T_c$) and the estimated curve ($E_c$) provided by the bio-motion generator. The ICC is determined from the variance of both curves ($T_c$) and ($E_c$) following the next equation:

$$\text{ICC} = \frac{\sigma_{E_c}^2}{(\sigma_{T_c}^2 + \sigma_{E_c}^2)}; \qquad (7)$$

On the other hand, SEM represents the existent difference between observed ($T_c$) and estimated curves ($E_c$) determined with the bio-motion generator, and provide an indication of the real magnitude of the error.

$$\text{SEM} = \sigma_c \cdot \sqrt{(1 - \text{ICC})}; \qquad (8)$$

Where $\sigma_c$ is the combined standard deviation of the true scores ($T_c$) and observed scores ($E_c$). And $S_E$ is the combined standard deviation of the true scores and observed scores.

We have obtained the SEM for each pair of true and predicted angles for the three spatial directions in all the joints that form the human model. For that reason, we have represented the SEM by its descriptive statistics (mean, std., 5-percentile and 95-percentile).

## RESULTS

### Parallel analysis

The results of the PA (Fig. 2) have been obtained with the explained variance of the main components extracted from the original data and the same obtained from randomized

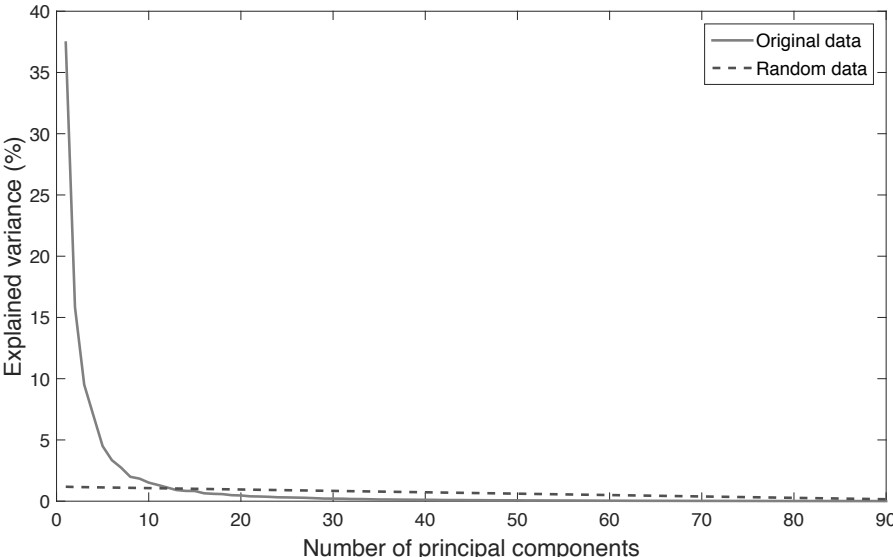

**Figure 2** Relation between the explained variance and the number of principal components extracted from the database and from randomized data.

data. The intersection point of both curves indicates the optimal number of components to extract from the PCA. The original number of dimensions was 72 (3 related to the pelvis translation + 69 related to the body segments orientation). The results of the PA recommend retaining the first 12 eigenvalues, which explain the 88.16% of the total variance. Thus, the PCA allowed a percentage of data reduction of 83%, from 72 variables to 12 weighed components.

## Regression model

As it has been explained in the methodology, the regression model consists of two parts, the first including the anthropometrical data (PLS) and the second the gender (LRM). The dependent variables of the PLS are the scores of the first 12 principal components (PC) of the kinematical running motion. Therefore, they are uncorrelated and the optimal number of PLS components are separately determined for each PC score (PC 1... PC 12) according to its adjusted $R^2$ plot (Fig. 3). PLS components are retained until their $R^2$ curve exhibits a decrease or a non-significant increase. Thus, for instance, two PLS components are retained for PC 1, whereas no components are considered for PC 7 and PC 9. Notice that for those PC with 0 retained components, the PLS model provides their mean value as output. This way, the motion information associated to those PC which is provided by the PLS model is the average motion.

With regard to the LRM, which analyses the influence of gender in the kinematics of running, the PCA scores which are significantly affected by gender are PC3, PC7, PC8, PC9 and PC11 (Table 2). The prediction obtained in the first step of the model is improved by the influence of gender on these PC. PC 7 and PC 9 are only affected by gender, since their number of retained PLS components was 0.

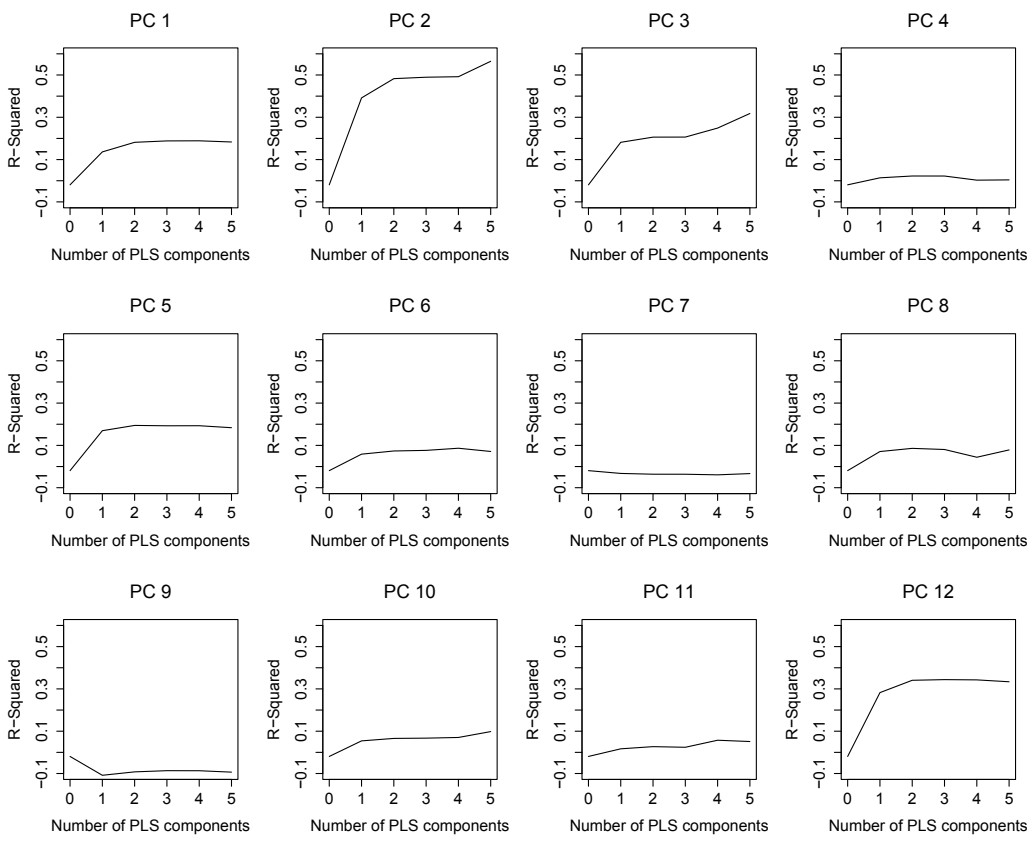

**Figure 3** Leave-one-out $R^2$ estimation plots for the PLS model.

## Validation of the bio-motion generator

The results of the reliability study, computed from the 90 observations and the same calculated by means of the leave-one-out technique, showed that the mean and standard deviation of ICC, was 0.91(0.04) with a 5 percentile of 0.829 and 95 percentile of 0.971. Only one subject exhibit an ICC lower than 0.8 in two observations (Fig. 4).

The SEM between the real and the predicted angles determined with the leave-one-out model showed a mean (std.) of 4.16° (6.80°), with a 5 percentile of 0.41° and 95 percentile of 14.23° (Fig. 5).

## DISCUSSION

In this paper we have demonstrated that the five-step methodology on which the bio-motion generator is based provides running motion models closely resembling the measurements obtained with real subjects. However, while the SEM study shows that the vast majority of errors detected between actual and predicted data of the bio-motion generator are less than 10°, there are a percentage of observations (8%) in which greater errors are observed. This can be explained because the model has been obtained from a small number of subjects—only 18—and therefore the bio-motion generator is not able to adjust the running specific characteristics of each corridor. Future work in this line of research

**Table 2** ANOVA table for the linear models.

|  |  | Df | Sum Sq | Mean Sq | *F*-value | Pr (>*F*) |  |
|---|---|---|---|---|---|---|---|
| PC1 | Gender | 1 | 214117 | 214117 | 1.092 | 0.298 | |
|  | Residuals | 103 | 20188135 | 196001 | | | |
| PC2 | Gender | 1 | 140256 | 140256 | 2.687 | 0.104 | |
|  | Residuals | 103 | 5377216 | 52206 | | | |
| PC3 | Gender | 1 | 259962 | 259962 | 5.592 | 0.0199 | * |
|  | Residuals | 103 | 4788029 | 46486 | | | |
| PC4 | Gender | 1 | 6242 | 6242 | 0.14 | 0.709 | |
|  | Residuals | 103 | 4591853 | 44581 | | | |
| PC5 | Gender | 1 | 79045 | 79045 | 3.434 | 0.0667 | |
|  | Residuals | 103 | 2370796 | 23017 | | | |
| PC6 | Gender | 1 | 2470 | 2470 | 0.125 | 0.724 | |
|  | Residuals | 103 | 2028601 | 19695 | | | |
| PC7 | Gender | 1 | 81973 | 81973 | 4.361 | 0.0392 | * |
|  | Residuals | 103 | 1935909 | 18795 | | | |
| PC8 | Gender | 1 | 88710 | 88710 | 7.979 | 0.00568 | ** |
|  | Residuals | 103 | 1145086 | 11117 | | | |
| PC9 | Gender | 1 | 62331 | 62331 | 4.962 | 0.0281 | * |
|  | Residuals | 103 | 1293890 | 12562 | | | |
| PC10 | Gender | 1 | 3246 | 3246 | 0.355 | 0.552 | |
|  | Residuals | 103 | 940560 | 9132 | | | |
| PC11 | Gender | 1 | 111297 | 111297 | 13.31 | 0.000417 | *** |
|  | Residuals | 103 | 861467 | 8364 | | | |
| PC12 | Gender | 1 | 1373 | 1373 | 0.29 | 0.591 | |
|  | Residuals | 103 | 487860 | 4737 | | | |

**Notes.**

Signification codes 0.001 '***'; 0.01 '**'; 0.05 '*'.

must be done to increase the database of real subjects measured and incorporate greater variability in anthropometric and performance characteristics.

The bio-motion generator is based on a methodology which comprises five steps. In the fourth step we tackle a dimensionality reduction based on PCA. This step is similar to that performed by *Troje (2002)*. However, there are some differences, as he obtained four main components that explain more than 98% of the variance and we have obtained 12 components explaining 88.16% of variance. The greater variability of our study is explained partly by the greater variability of the running against walking and on the other hand by the greater speed range in our study in relation to Troje, in which each subject could select a single comfortable walking speed. On the other hand, Troje made a second reduction of the dimensionality based on the simplicity of temporal behaviour of the walking components which could be modelled with pure sine functions with a scaled fundamental frequency. This approach was not valid for the motion of running, due to the fact that the 12 PCs of running cannot be modelled with a proportional frequency. This suggests that running is a more complex motion than walking in the sense that there does not exist a proportion between the frequency of oscillation of the different body segments.

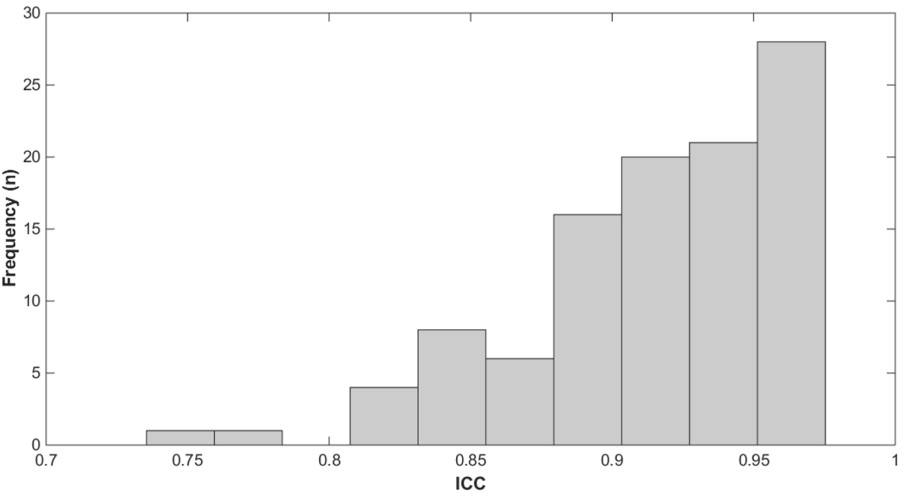

**Figure 4  Frequency histogram of the ICC.**

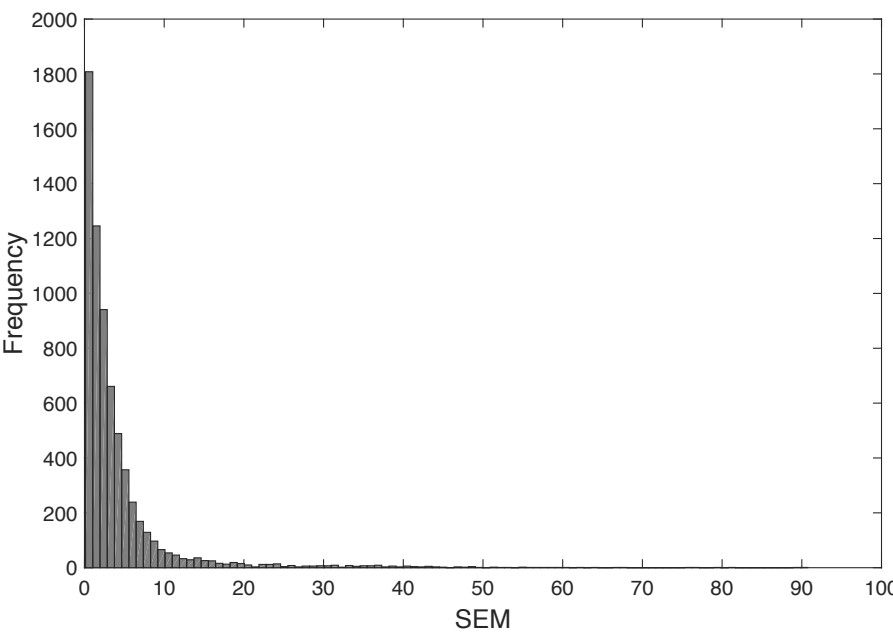

**Figure 5  Frequency histogram of the SEM.**

The fifth step of the methodology consists of a two-step linear regression which correlates a given list of 1D measurements with the PCA scores of movement. A linear regression technique has been used before to approximate motion models from a reduced marker set and estimate the remaining markers (*Liu et al., 2005*) or to model the motion-style and the spatio-temporal movement (*Torresani, Hackney & Bregler, 2006*). However, it has not been used before to synthesize new human motion directly from a set of anthropometrical and performance data. In this sense, it can be considered a real breakthrough in the field of synthesis of human motion.

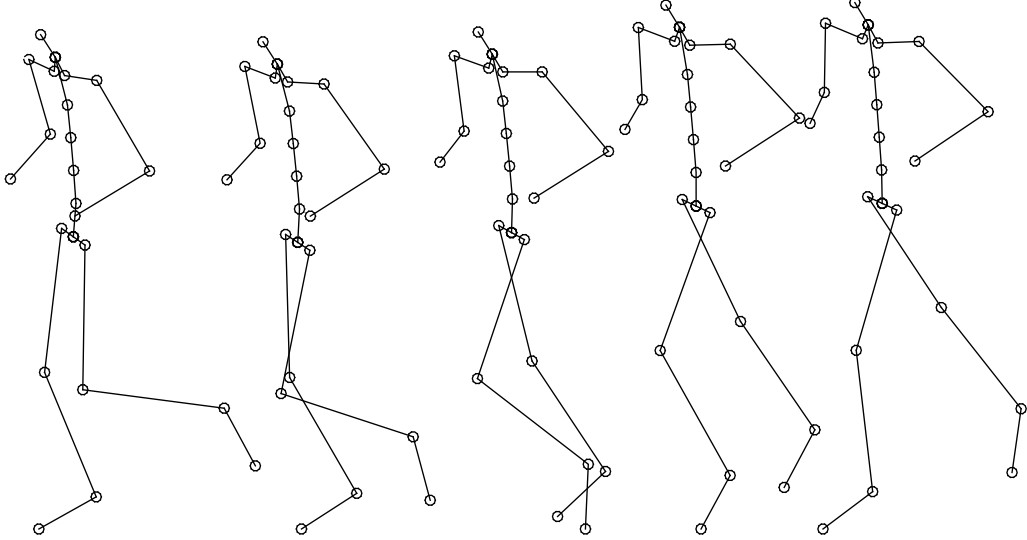

**Figure 6** **Reconstructed virtual biomechanical model** (skeleton+motion).

## CONCLUSIONS

The major contribution of this paper is a novel statiscal methodology for modelling human movements. The method described in this article has been developed and validated for running motion, but this same methodology could be used to synthesize other types of motion: walking, going up and down stairs, or even for sport movements such as: jumping, pedalling, golf swing and putting, etc.

Our work aims to provide a realistic motion to body shapes that can be developed with the methodology described in the work of *Ballester et al. (2014)*. Those body shapes could include an adjusted skeleton formed by a hierarchical set of interconnected joints and can be used to move the body shape with the required or desired motion provided by our methodology (Fig. 6). The integration of both methods will allow generating realistic avatars supplied with realistic motion from a set of adjustable and simple anthropometrical and performance data and without the need of the realization of new measurements.

A limitation of this study is the sample size. Further work needs to be done in order to validate with a broader sample of people. Notwithstanding this limitations, the findings suggest that the model is valid.

### Funding
The research for this paper was done within the EASY-IMP project (http://www.easy-imp.eu/) funded by the European Commission FP7.FoF.NMP.2013-5 Project 609078. The funders had no role in study design, data collection and analysis, decision to publish, or preparation of the manuscript.

### Grant Disclosures

The following grant information was disclosed by the authors:
European Commission: FP7.FoF.NMP.2013-5 Project 609078.

### Competing Interests

The authors declare there are no competing interests.

### Author Contributions

- José María Baydal-Bertomeu conceived and designed the experiments, performed the experiments, analyzed the data, wrote the paper, prepared figures and/or tables, performed the computation work.
- Juan Vicente Juan V. Durá-Gil conceived and designed the experiments, performed the experiments, analyzed the data, wrote the paper, prepared figures and/or tables, reviewed drafts of the paper.
- Ana Piérola-Orcero analyzed the data, wrote the paper, prepared figures and/or tables, performed the computation work.
- Eduardo Parrilla Bernabé prepared figures and/or tables, performed the computation work.
- Alfredo Ballester and Sandra Alemany-Mut reviewed drafts of the paper.

### Ethics

The following information was supplied relating to ethical approvals (i.e., approving body and any reference numbers):

Ethics Committee. Universitat Politècnica of Valencia.

### Data Availability

The raw data has been supplied as Supplementary File.

### Supplemental Information

Supplemental information for this article can be found online at http://dx.doi.org/10.7717/peerj-cs.102#supplemental-information.

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
