# Peer review of "A PCA-based bio-motion generator to synthesize new patterns of human running"

_PeerJ Computer Science, doi:10.7717/peerj-cs.102_

## Round 0.1 · original submission · Major Revisions

Both reviewers have serious concerns about the paper so we cannot accept it in its current form. I would encourage the authors to prepare a new version addressing all the corrections proposed, including a revision letter explaining how you addressed these corrections.

Reviewer 1 ·

Basic reporting

The paper describes an approach to synthesize running motions from examples by decomposing the motion into a set of variables such as gender, age,height, etc.
In general the paper addresses an interesting problem, the experiments are well done and results are interesting. However, the technical description of the paper does not meet the necessary quality for a journal publication. The authors are encouraged to rewrite the paper in a more mathematically formal way since the work is interesting.

Below is a more detailed summary of my concerns:

1) The dataset is fine but contains almost no variability on the data the researchers want to measure. The researchers indicate that one of their goals is to
"determine a statistical model capable of synthesizing new
realistic running motion from a set of desired data: age, gender, height, body mass index (BMI)
and velocity."

However, the limited number of people in their experiments does not allow to do that. Which are the height and BMI of the subjects included? It is not indicated.
Also, according to the tables of age, there is little variation in that dimension to obtain reliable models.

2) line 170. They indicate 3200 columns. However, it is not clear why. They should clarify on that point. If they are using 20 joints, it makes roughtly 60 variables for the whole model in a given time instant. So, it makes around 50 measures, which at a frame rate of 120Hz, is no more than half second. Is that right?
In any case, the authors must indicate much more clearly the number of variables of their model, a picture of it and where the 3200 columns come from.

3) This links to the Mathematical procedure in line 139. It would be much more technical to have steps 1-3 be somehow graphically shown with some figures.

In general there is a lack of mathematical formalization of the procedure.

4) The explanation of the PCA dimensionality reduction is not very technical.

5) The term bio-motion generator is randomly introduced in the text. First in the title, but it does not appear again until line 173 (near the middle of the paper).

6) The authors apply a PLS approach coupled with a LRM to do the inference from age,gender,etc to movement. However, the authors should provide a more complete explanation of these techniques.

7) line 208. The authors should explicitly indicate the domain of x (line 208), i.e., what is the size of a_i. Also, which is the domain of variable "gender". In general a more technical and formal description of all variables is required.
line 220, coefficients alpha and beta, have 12 components. Why 12?

8) Equations 6,7 are not way of calculating values (line 245), they are representation of the curve values.

9) ICC should be an equation, though, instead of just text (line 249)

10) The value Se in Eq 8, is not clearly indicated how it is computed.

Experimental design

No Comments

Validity of the findings

No Comments

Reviewer 2 ·

Basic reporting

The english language written in the paper is OK. The paper is easy to read and to understand

The background of the problem is well shown into introduction section and the references used are well chosen mixing old relevant and recent references on the problem.

In general, the structure of the paper conforms the recommendations of Perj Standard. The authors include a section of methods. The section of Materials is named by the authors "instrumentation" and it include reasonings on the methodology used.
This organization can be improved including all expositions on methodology in section Methods that could be named Materials & Methods.

Experimental design

The authors propose to use PCA to selection of main features on movement of an human to generate new movements according with other variables (age and gender) that have been not considered for this problem. The idea is not very original because the authors propose a similar methodology similar to published by Vinué et al., 2014 (see paper lines 88 -95).
The methodology proposed is OK but, in my opinion, there are two relevant problems not considered by the authors. This problem is related with validation of the proposed methodology.

Validity of the findings

First problem: The authors use a commercial system based on 17 inertial sensors but there is not any information on the previous validation system. Without this prerequisite all data may not be valid.

Second problema: Because the main objective of work is to predict movements of humans with similar features but different age and/or gender, where is the real data sample to confirm the results obtanided?

Additional comments

In my opinion the idea is well but to confirm the conclusions of the paper is needed a major effort to obtain a real sample of humans with similar features of body and differents age and gender, and it is not easy.

Also, the authors should have a effort to clarify the differences between the proposed methodology and the methodology proposed by Vinué et al., 2014 to judge the originality of the work

---

## Round 0.2 · accepted · Accept

As this new version has addressed most of the reviewer's comments I believe the manuscript is ready for publication.

Reviewer 1 ·

Basic reporting

The authors have made some changes on the paper considering our review. Some aspect of the model have been clarified. Nevertheless, my main concerns are two. First, the dataset contains very few people for a model with so many parameters, and the variability is very reduced (Table 1). Second, there is no comparison with existing techniques

Experimental design

No comparison with other techniques.

Validity of the findings

Because reduced size of the dataset, the validity of the findings can not be assesed.

Reviewer 2 ·

Basic reporting

The authors have managed well the answers corrections according to my previous suggestions. So my opinion is favorable to accept the paper

Experimental design

With limitations but is OK

Validity of the findings

No Comments

Additional comments

No Comments